# Overexpression of the Transcription Factor *GmbZIP60* Increases Salt and Drought Tolerance in Soybean (*Glycine max*)

**DOI:** 10.3390/ijms26073455

**Published:** 2025-04-07

**Authors:** Mengnan Chai, Fan Yang, Shuping Cai, Tingyu Liu, Xiaoyuan Xu, Youmei Huang, Xinpeng Xi, Jiahong Yang, Zhuangyuan Cao, Ling Sun, Danlin Dou, Xunlian Fang, Maokai Yan, Hanyang Cai

**Affiliations:** 1Fujian Provincial Key Laboratory of Haixia Applied Plant Systems Biology, State Key Laboratory of Ecological Pest Control for Fujian and Taiwan Crops, Haixia Institute of Science and Technology, School of Future Technology, College of Life Sciences, Fujian Agriculture and Forestry University, Fuzhou 350000, China; chaimengnan1@163.com (M.C.); atlas_yf@163.com (F.Y.); cai89744890@outlook.com (S.C.); 15904444973@163.com (T.L.); xuxiaoyuan2025@163.com (X.X.); hym9995@163.com (Y.H.); 18235795238@163.com (X.X.); yangjiahong1224@163.com (J.Y.); caozhuang09@163.com (Z.C.); sunling1998@163.com (L.S.); doudanlin@163.com (D.D.); 1589419619fxl@sina.com (X.F.); 2State Key Laboratory for Conservation and Utilization of Subtropical Agro-Bioresources, Guangxi Key Laboratory of Sugarcane Biology, College of Agriculture, Guangxi University, Nanning 530004, China; yanmaokai123@163.com

**Keywords:** *GmbZIP60*, soybean, transcription factor, salt, drought

## Abstract

The regulation of downstream responsive genes by transcription factors (TFs) is a critical step in the stress response system of plants. While bZIP transcription factors are known to play important roles in stress reactions, their functional characterization in soybeans remains limited. Here, we identified a soybean bZIP gene, *GmbZIP60*, which encodes a protein containing a typical bZIP domain with a basic region and a leucine zipper region. Subcellular localization studies confirmed that *GmbZIP60* is localized in the nucleus. Expression analysis demonstrated that *GmbZIP60* is induced by salt stress, drought stress, and various plant hormone treatments, including abscisic acid (ABA), ethylene (ETH), and methyl jasmonate acid (MeJA). Overexpressing *GmbZIP60* (*OE-GmbZIP60*) in transgenic soybean and rice enhanced tolerance to both salt and drought stresses. Quantitative real-time polymerase chain reaction (qRT-PCR) analysis indicated that the expression levels of abiotic stress-responsive genes were significantly higher in transgenic plants than in wild-type (WT) plants under stress conditions. Chromatin immunoprecipitation-qPCR (ChIP-qPCR) analysis further confirmed that GmbZIP60 directly binds to the promoters of abiotic stress-related genes induced by ABA, ETH, JA, and salicylic acid (SA). Overall, these findings revealed *GmbZIP60* as a positive regulator of salt and drought stress tolerance.

## 1. Introduction

Soybean (*Glycine max*) is a globally significant crop extensively cultivated across various regions [1,2]. It is rich in protein, making it a primary source of plant-based protein worldwide [3,4,5]. Soybean by-products, such as soybean meal, serve as an important protein source in animal feed [6]. Besides providing food, soybean cultivation contributes significantly to nutrient cycling and environmental sustainability [7]. Soybeans are extensively cultivated in regions throughout the Americas, Asia, and Africa, where they are frequently exposed to extreme environmental stresses, such as drought and soil salinity [8], due to substantial climatic variability. To adapt to these challenging conditions, soybeans have developed complex signaling transduction pathways with transcription factors playing a crucial role in regulating stress response.

Basic leucine zipper (bZIP) transcription factors are defined by their structural domains, comprising a basic region and a leucine zipper region [9]. The basic region, highly conserved and located at the N-terminus of the bZIP domain, contains 16–20 basic amino acid residues responsible for DNA binding, with the core consensus sequence being ACGT. The leucine zipper region, situated at the C-terminus, consists of one or more heptad repeats, each containing a leucine or other hydrophobic residue. These residues facilitate the dimerization of two bZIP transcription factors through hydrophobic interactions, resulting in the formation of a supercoiled structure [10,11].

bZIP transcription factors are essential for various biological processes in soybeans, including growth, development, stress responses, and the regulation of secondary metabolism [12,13]. For instance, overexpression of *GmbZIP2* has been demonstrated to enhance drought tolerance by reducing malondialdehyde (MDA) levels and increasing the activity of antioxidant enzymes such as catalase (CAT), superoxide dismutase (SOD), and peroxidase (POD), thereby improving higher survival rate under drought conditions [14]. Additionally, genes such as *GmTRAB1* and *GmbZIP2* are involved in regulating plant hormone signaling pathways, including ABA, ETH, and JA, which are crucial for plant responses to abiotic stresses like drought and salinity [15,16]. Similarly, the AtbZIP28 transcription factor mediates the heat stress response by regulating HSFA2, highlighting the role of bZIP factors in enhancing stress resistance by regulating genes related to stress responses [17]. In pepper, the bZIP transcription factor CaADBZ1 plays a significant role in drought stress response by modulating plant sensitivity to ABA [18]. Equally, overexpression of *GmbZIP15* enhances disease resistance in soybean plants against pathogens such as *Sclerotinia sclerotiorum* and *Phytophthora sojae* [19]. However, *GmbZIP15* overexpression simultaneously reduces soybean tolerance to abiotic stresses [14].

In addition to stress adaptation, bZIP family members modulate plant growth and development. For example, CmbZIP19 directly binds to the ZDRE-like element in the promoter region of CmDWF1, repressing its expression and consequently inhibiting the synthesis of brassinosteroids, which suppresses the elongation of lateral buds in chrysanthemum [20]. Likewise, the bZIP transcription factor AtHY5 regulates hypocotyl growth and pigment accumulation in a light-dependent manner in Arabidopsis [21]. This further demonstrates the regulatory function of bZIP transcription factors in plant growth and development.

In our previous studies, the expression of the GmbZIP60 gene, a member of the bZIP transcription factor family, was markedly upregulated in wild-type soybean Wil-liam82 under drought and flood stresses. This finding indicates that GmbZIP60 likely plays a pivotal role in the plant’s response to different abiotic stresses, thereby prompting our in-depth investigation into this gene [22]. In this study, we identified and cloned the *GmbZIP60* gene, which is localized in the nucleus. Expression patterns analysis revealed that *GmbZIP60* is induced by both salt and drought stress. Moreover, overexpression of *GmbZIP60* significantly enhanced salt and drought tolerance in transgenic plants compared to wild-type (WT). Additionally, this study reveals that the GmbZIP60 transcription factor exhibits cross-species functional conservation, conferring salt and drought stress tolerance not only in soybeans but also in rice. These findings highlight the crucial role of *GmbZIP60* in mediating responses to multiple abiotic stresses. Overall, our results suggest that the *GmbZIP60* is not only essential for enhancing soybean resilience to environmental stress but may also contribute to promoting growth and yield under prolonged stress conditions.

## 2. Results

### 2.1. Sequence and Domain Analyses of GmbZIP60

*GmbZIP60* cDNA is 450 bp in length (Appendix A) and encodes a bZIP-structured protein (Appendix A) with a relative molecular mass of 16.92 kDa and a theoretical isoelectric point (pI) of 4.79. The sequence of GmbZIP60 (Glyma.02G012700) and its homologous protein were downloaded from Phytozome13. Multiple sequence alignments revealed that GmbZIP60 shares high sequence similarity with GmbZIP152 (Glyma.19G216200), AtbZIP53 (AT3G62420), AtbZIP44 (AT1G75390), and OsbZIPOBF1 (LOC_Os12g37410). GmbZIP60 and its homologous proteins contained a nuclear localization signal (NLS) and two conserved domains typical of the bZIP gene family: the basic region and the leucine zipper region (Appendix A).

### 2.2. Subcellular Localization of GmbZIP60

To explore the subcellular localization of GmbZIP60, a fusion construct, *35S::GmbZIP60::GFP* was generated by fusing the *GmbZIP60* coding sequence with a green fluorescent protein (GFP). This recombinant vector was then transformed into *Nicotiana benthamiana* leaves via agroinfiltration, with the empty vector (*35S::GFP*) serving as the control. The results showed that *35S::GmbZIP60::GFP* fluorescence was exclusively localized in the nucleus, while *35S::GFP* fluorescence was detected in both the cytoplasm and the nucleus (Figure 1). These results indicate that GmbZIP60 localizes in the nucleus.

### 2.3. Expression Patterns of GmbZIP60 in Response to Various Treatments

Several studies have reported that bZIP transcription factors regulate plant responses to abiotic stresses. To investigate whether GmbZIP60 is involved in abiotic stress response, two-week-old soybean plants were exposed to various stress treatments, and the leaves were collected for qRT-PCR analysis. Following salt treatment, *GmbZIP60* expression gradually increased, reaching the maximum level at 12 h (Figure 2A). Under drought treatment, *GmbZIP60* expression peaked at 6 h but significantly decreased at 12 h (Figure 2B).

ABA, ETH, and MeJA play crucial roles in plant stress responses. ABA has been shown to facilitate the maintenance of water balance and antioxidant capacity by modulating stomatal closure and antioxidant enzyme activities. The sensitivity of plants to ABA is an indicator of their stress response. ETH contributes to the regulation of water balance under salt and drought stress by modulating proline metabolism. MeJA has been demonstrated to enhance plant resistance under various stresses by regulating the expression of defense-related genes and the antioxidant system. Furthermore, the sensitivity of plants to ABA, ETH, and MeJA is also influenced by the role of the GmbZIP60 in stress response. ABA and ETH treatment resulted in a maximal *GmbZIP60* expression level at 6 h, followed by a gradual decrease, reaching a minimum at 24 h (Figure 2C,E). In addition, we observed a significant upregulation of *GmbZIP60* expression, with a maximal response occurring at 12 h after MeJA treatment (Figure 2D).

To confirm the results of qRT-PCR analysis, a β-Glucuronidase (GUS) reporter gene driven by the *GmbZIP60* promoter was used to generate the *pGmbZIP60::GUS* construct, which was subsequently transformed into *Arabidopsis* via *Agrobacterium tumefaciens*. WT and *pGmbZIP60::GUS* transgenic *Arabidopsis* seedlings were grown on 1/2 MS medium for 7 days and then transferred to 1/2 MS medium supplemented with 150 mM NaCl (salt stress), 400 mM mannitol (drought stress), 1.0 μM ABA, 150 μM MeJA, or 400 μM ETH for 6 to 24 h (Figure 2F–V). Untreated seedlings served as controls. Under control conditions, *GmbZIP60* expression was undatable in the WT but exhibited weak expression in the leaves of *pGmbZIP60::GUS* transgenic *Arabidopsis* seedlings (Figure 2F,G). Following stress treatments, *GmbZIP60* expression was strongly induced at 12 h after NaCl and MeJA treatments (Figure 2I,R) and at 6 h after drought, ABA, and ETH treatments (Figure 2I,N,K). The GUS activity was consistent with the qRT-PCR results (Figure 2A–E).

### 2.4. Overexpression of GmbZIP60 Enhances the Resistance of Transgenic Soybean Plants to Salt and Drought Stresses

To further investigate the role of *GmbZIP60* in response to salt and drought stresses, seedlings of *Arabidopsis* and two independent transgenic lines overexpressing *GmbZIP60* (*OE-16* and *OE-18*) were exposed to stress conditions (Appendix A). As a control, WT, *OE-16*, and *OE-18* seedlings were grown on 1/2 MS medium. For salt stress, 1/2 MS medium was supplemented with 100 mM and 150 mM NaCl, while drought stress was simulated using 200 mM and 300 mM mannitol. In the control group, no significant differences were observed in root length and fresh weight among WT, *OE*-16, and *OE*-18 seedlings. Similarly, under 200 mM and 300 mM mannitol treatment root length and fresh weight of *OE*-16 and *OE*-18 seedlings showed no significant differences compared to WT (Appendix A). However, under salt stress conditions (100 mM and 150 mM NaCl) root length and fresh weight of *OE*-16 and *OE*-18 seedlings significantly increased compared to WT (Appendix A). Additionally, *OE*-16, *OE*-18, and WT seedlings were treated with different concentrations of ABA, ETH, and MeJA (Appendix A). The results showed that the root length and fresh weight of *OE*-16 and *OE*-18 were significantly reduced compared to WT, suggesting that the transgenic plants exhibit higher sensitivity to these hormones. To further investigate the role of *GmbZIP60* under abiotic stress conditions, transgenic soybean lines overexpressing *GmbZIP60* (*OE-GmbZIP60*) were generated. Two lines, *OE-12* and *OE-44*, were selected for further analysis based on their higher expression levels (Appendix A). Seedlings of WT and the *OE-GmbZIP60* lines were subjected to 150 mM NaCl or 350 mM mannitol treatments for 20 days. Under control conditions, the growth of *OE-12* and *OE-44* seedlings was comparable to that of WT, although WT exhibited slightly greater plant height. Following salt treatment, the leaves of WT turned white, and the plants exhibited severe wilting, nearing fatality. In contrast, *OE-12* and *OE-44* soybean seedlings exhibited only slight wilting with yellow leaves and maintained growth activity. Similarly, WT plants displayed more severe wilting under drought stress, whereas *OE-16* and *OE-18* seedlings showed only slight wilting, with their stems remaining more vigorous than those of WT (Figure 3A). Stress-tolerant plants can maintain a relatively high photosynthetic rate and water content under adverse conditions. They mitigate the impact of stress on growth and metabolism by regulating stomatal aperture and optimizing water use efficiency [23,24]. Consistent with the observed phenotypic differences, the photosynthetic activity of different plants also varied under different stress conditions. Under control conditions, the fluorescence parameters of photosynthetic activity in *OE*-12 and *OE*-44 seedlings showed no significant difference compared to the wild type (WT). After treatment with salt and drought stresses, the photosynthetic activity of WT, *OE*-12, and *OE-*44 decreased compared to the control group. However, under salt and drought stresses, the photosynthetic activity of *OE*-12 and *OE*-44 was significantly higher than that of WT (Figure 3B). Additionally, we measured the water content of the plants. Under salt and drought stresses, the water content of *OE*-12 and *OE*-44 was significantly higher than that of WT (Figure 3C). These results suggest that overexpression of *GmbZIP60* enhances the resistance of soybeans to salt and drought stresses.

### 2.5. Overexpression of GmbZIP60 Enhances the Resistance of Transgenic Rice Plants to Salt and Drought Stresses

Rice is a crucial food crop that typically grows in waterlogged conditions, making it particularly vulnerable to various abiotic stresses, including salt and drought. Therefore, enhancing the resilience of rice to this stress is of great importance.

To investigate the role of *GmbZIP60* in rice, two transgenic rice lines overexpressing *GmbZIP60* (*OE-2* and *OE-5*) were generated (Appendix A) and subjected to drought and salt stress for further analysis. Salt stress was simulated using 100 mM and 150 mM NaCl, while drought stress was induced using 250 mM and 350 mM mannitol. After 4 days of treatment, the bud lengths of the *OE-GmbZIP60* transgenic rice plants were significantly longer than those of WT (Figure 4A,C). After 10 days of treatment, the leaf and root length of *OE-2* were conspicuously increased compared with WT under both 150 mM NaCl and 350 mM mannitol treatments (Figure 4B,D,E). These results suggest that overexpression of *GmbZIP60* enhances salt and drought tolerance in rice.

### 2.6. Expression Analysis of Stress-Related Genes in OE-GmbZIP60 Transgenic Rice and Soybean Under Salt and Drought Stress

To investigate increased resistance of soybean plants overexpressing *GmbZIP60* to salt and drought stresses, qRT-PCR was performed to analyze the expression levels of soybean genes associated with salt stress, including *GmUBC*, *GmWRKY111*, *GmWRKY33* [25], and *GmWRKY28* (Figure 5A–D), as well as drought stress-related genes such as *GmUBC* [14,26], *GmDREBa* [27], *GmDREBb* [27], and *GmMYB118* [28] (Figure 5E–H) in *OE-12*, *OE-44*, and WT plants. The results indicated that the expression levels of these stress-related genes were significantly higher in *OE-12* and *OE-44* compared to WT (Figure 5). Additionally, salt and drought treatments lead to increased gene expression in both WT and *OE-GmbZIP60* transgenic soybean plants, with a more pronounced upregulation observed in the *OE-12* and *OE-44* (Figure 5). These findings suggest that overexpression of *GmbZIP60* enhances soybean tolerance to salt and drought stress by upregulating stress response genes.

We further examined the expression levels of four abiotic stress response genes in rice, including *OsDREB2A* [29], *OsDREB2B* [29], *OsRD29A* [30], and *OsLEA3* [31]. This analysis was conducted using two-week-old WT, *OE-2*, and *OE-5* under control, salt, and drought conditions. qRT-PCR was performed to monitor gene expression at 0 h, 24 h, and 48 h after salt and drought stress treatments. The results revealed that the expression levels of these stress-related genes were significantly higher in the *OE-2* and *OE-5* compared to WT under both stress conditions (Figure 6). Following salt stress treatment, *OsDREB2A*, *OsDREB2B*, *OsRD29A*, and *OsLEA3* were all upregulated in WT, *OE-2,* and *OE-5*, but the level of induction was much greater in the *OE-2* and *OE-5* (Figure 6A–D). Similarly, after drought stress treatment, *OE-2* and *OE-5* exhibited higher expression levels of these four stress-related genes compared to WT plants (Figure 6E–H). These results suggest that the increased expression of stress-responsive genes contributes to the enhanced tolerance to salt and drought stress in *OE-GmbZIP60* transgenic rice plants.

### 2.7. Identification of Target Genes for GmbZIP60

To explore the molecular pathways through which *GmbZIP60* regulates stress tolerance, Chromatin immunoprecipitation (ChIP) was conducted to identify potential target genes of GmbZIP60. Two-week-old *OE-12* transgenic soybean leaves were collected for the ChIP experiments, followed by ChIP-qPCR to detect DNA fragments bound by GmbZIP60. The results revealed that GmbZIP60 was enriched in the promoters of several abiotic stress-related genes, including *GmABI5*, *GmERD1*, *GmETR2*, *GmRD22*, *GmDERBb*, *GmEIN2*, *GmPR2*, and *GmBIP*, with multiple binding sites identified for GmbZIP60 in the promoters of these genes (Figure 7). Notably, these genes are associated with various hormone signaling pathways: *GmABI5* and *GmRD22* are related to ABA signaling, *GmETR2* and *GmEIN2* relate to ETH signaling, *GmRD22* related to JA signaling, and *GmPR2* is involved in SA signaling. These findings suggest that overexpression of *GmbZIP60* enhances abiotic stress by increasing the transcription of genes involved in abiotic stress and multiple hormone signaling pathways.

## 3. Discussion

Soybean is rich in protein and oil, making it valuable in food, feed, and industrial applications [32]. Additionally, soybean cultivation contributes to agricultural sustainability through nitrogen fixation and soil improvement [33]. However, with climate change and environmental degradation, abiotic stresses such as drought and salt stress have become major factors limiting soybean yield and quality [34,35]. Therefore, identifying and studying key genes involved in abiotic stress responses is essential for improving soybean resilience. bZIP transcription factors play critical roles in various biological processes, such as plant growth, development, and seed maturation [36]. For example, *AtbZIP30*, expressed in meristematic tissues and functions as a negative regulator of plant growth and reproductive development of *Arabidopsis* [37]. Furthermore, bZIP transcription factors are essential in both biotic and abiotic stress responses [38,39,40]. In this study, we isolated and cloned the *GmbZIP60* gene from the soybean genome.

Sequence alignment revealed that GmbZIP60 contained a highly conserved basic region and leucine zipper region, sharing high sequence similarity with GmbZIP152 from soybean, AtbZIP44 and AtbZIP53 from *Arabidopsis*, and OsOBF1 from rice (Appendix A). Several studies have shown that these homologs play roles in stress response: *GmbZIP152* was significantly induced by salt, drought, and heavy metal stress [41]; *AtbZIP44* positively regulated the response of *Arabidopsis* to Fe deficiency stress by interacting with AtMYB10 and AtMYB72 [42]; *AtbZIP53* is involved in salt and starvation-mediated stress response [43,44]; and *OsOBF1* is induced by low temperature [45]. Consistent with these findings, we demonstrated that *GmbZIP60* expression is upregulated under salt and drought stress treatments (Figure 2A,B). Additionally, GUS activity in *pGmbZIP60::GUS* transgenic plants was also found to be elevated in response to salt and drought stress, suggesting that *GmbZIP60* is involved in response to these environmental stressors.

Accumulating evidence suggests that bZIP transcription factors play significant roles in regulating tolerance to abiotic stress. Some bZIP transcription factors can form homodimers with other bZIP members to jointly regulate metabolic pathways, thereby contributing to plant stress responses. For example, in Arabidopsis, the bZIP transcription factor AtbZIP53 from the S subfamily cannot form functional homodimers with other members of its subfamily, but it can form heterodimers with AtbZIP10 from the C subfamily to regulate proline metabolism [46]. Moreover, bZIP transcription factors can also interact with certain proteins, such as BLADE-ON-PETIOLE (BOP), to collectively regulate plant development [47]. For instance, transgenic maize plants overexpressing *ZmbZIP4* exhibit improved tolerance to salt and drought stresses [48]. Similarly, the transcription factor OsbZIP23 in rice enhances drought tolerance by regulating drought response genes in conjunction with the H3K4me3 histone modification [49]. Consistent with previous findings, the present study demonstrated that overexpression of *GmbZIP60* in *Arabidopsis*, soybean, and rice significantly improved stress response. Under salt conditions, the root length and fresh weight of *OE-GmbZIP60* transgenic *Arabidopsis* plants were significantly increased compared with those in WT (Appendix A). Similarly, *OE-GmbZIP60* transgenic soybeans plants exhibited stronger growth under salt and drought conditions (Figure 3), while *OE-GmbZIP60* transgenic rice plants displayed longer buds, leaves, and roots compared to WT plants under stress conditions (Figure 4). Additionally, no significant improvement in drought resistance was observed in *OE-GmbZIP60* transgenic *Arabidopsis* plants, possibly due to the limited role of *GmbZIP60* in drought stress in *Arabidopsis*. However, rice, which is more sensitive to drought due to its unique cultivation method, exhibited increased drought tolerance in *OE-GmbZIP60* transgenic lines. These results indicate that overexpression of *GmbZIP60* enhances plant growth and development under salt and drought stresses.

To explore the molecular pathways through which *GmbZIP60* regulates stress tolerance, qRT-PCR was performed to analyze the expression level of stress-related genes in *OE-GmbZIP60* transgenic and WT plants. In our study, the expression levels of various stress-responsive genes, including *AtABI5*, *AtRD29B*, *AtWRKY26*, *AtABA2*, *AtCOR6-6*, and *AtSTZ* in *Arabidopsis* (Appendix A), *OsDREB2A*, *OsDREB2B*, *OsRD29A*, and *OsLEA3* in rice (Figure 5), *GmUBC*, *GmWRKY111*, *GmWRKY33*, *GmWRKY28*, *GmDREBa*, *GmDREBb*, and GmMYB118 in soybean were significantly upregulated in *OE-GmbZIP60* transgenic plants under stress conditions (Figure 6). Interestingly, we found that even under non-stress conditions, the expression levels of certain genes varied among different overexpression lines. Under drought and salt stress, different plants exhibited varying degrees of stress response, suggesting that the expression levels of GmbZIP60 in *GmbZIP60-OE-12* and *GmbZIP60-OE-44* might differ. Although qRT-PCR results showed that the expression level of *GmbZIP60* in *GmbZIP60-OE-12* was higher than in *GmbZIP60-OE-44*, the actual protein levels may have reached a threshold beyond which further accumulation was not possible. As a result, the protein levels of GmbZIP60 in *GmbZIP60-OE-44* and *GmbZIP60-OE-12* could be comparable. These findings indicated that *GmbZIP60* enhances plant stress tolerance by modulating the expression of stress-related genes.

Plant hormones like ABA, JA, and ETH mediate plant responses to abiotic stresses. Under stress conditions, endogenous ABA rapidly accumulates, activating the expression of stress-responsive genes and triggering a broad range of physiological responses [50,51,52]. For example, *TabZIP60* regulates plant tolerance to drought, salt, and freezing stresses by participating in the ABA signaling pathway [53]. JA plays a central role in the mediation of plant responses and defense to abiotic stresses and has been extensively studied [54]. OsJAZ9 is the repressor of JA in rice and significantly increases drought tolerance by modulating JA signaling [55,56]. ETH is essential for regulating plant growth, development, and stress responses, and its biosynthesis significantly enhances salt stress tolerance [57,58]. In our results, the expression of *GmbZIP60* was activated by diverse hormones, including ABA, ETH, and JA (Figure 2C–E), and the GUS activity of *pGmbZIP60::GUS* was also induced by the hormones (Figure 2N–V). The *OE-GmbZIP60* transgenic *Arabidopsis* showed a response to ABA, ETH, and SA (Appendix A). These findings suggest that *GmbZIP60* may be involved in abiotic stress response through different hormone signaling pathways. To further investigate the involvement of *GmbZIP60* in hormone signaling, the expression level of several hormone-related genes in *OE-GmbZIP60* transgenic *Arabidopsis* plants was analyzed. The expression of *AtABI5*, *AtABA2*, and *AtCOR6-6* was significantly higher in *OE-GmbZIP60* transgenic *Arabidopsis* plants than in WT plants under stress conditions (Appendix A). ChIP-qPCR analysis showed that GmbZIP60 directly binds to the promoter of several hormone-associated stress-related genes (Figure 7). These findings further confirm that *GmbZIP60* plays a crucial role in regulating abiotic tolerance through hormone signaling pathways.

Our results showed that the overexpression of *GmbZIP60* significantly enhances the tolerance of transgenic *Arabidopsis* to salt stress. Additionally, it plays a crucial role in increasing the resilience of transgenic rice and soybean to both salt and drought stress. This finding suggests that *GmbZIP60* is a transcription factor with conserved functions across various plant species, regulating stress response by regulating genes associated with plant hormone signaling and abiotic stress pathways (Figure 8). This study provides a solid theoretical foundation for the functional characterization of *GmbZIP60* and highlights its potential application in developing stress-resilient crops. However, further in-depth investigation is essential to comprehensively understand the intricate molecular mechanisms of *GmbZIP60* under stress conditions.

## 4. Materials and Methods

### 4.1. Plant Growth Conditions

The wild-type plants in this study included *Arabidopsis* (Columbia, Col-0), soybean (William 82), and rice (Zhonghua 11, ZH11). Growth conditions for Arabidopsis thaliana and soybean are described in a previous study [34]. Growth conditions of rice were maintained at 26–28 °C with 50% humidity under a photoperiod of 14 h of light and 10 h of darkness.

### 4.2. GmbZIP60 Gene Isolation, Vector Construction

Total RNA was extracted from the leaves of soybean variety William 82 using an RNA extraction kit from Omega Bio-Tek (Shanghai, China). cDNA was synthesized with PrimerScript™ RTase from TAKARA (Dalian, China). Specific primers (Appendix A) were utilized for the PCR amplification of *GmbZIP60*. The PCR products were then cloned into the vectors pGWB533 and pGWB605.

### 4.3. Arabidopsis, Rice, and Soybean Transformation

Genetic transformations were performed using *Agrobacterium tumefaciens*, including the floral dip method for *Arabidopsis* [41], the cotyledonary node method for soybean [59,60], and the callus infection method for rice [61].

### 4.4. Stress Tolerance Assays and Measurements of Physiological Indices

To study the expression profile of *GmbZIP60* in *Arabidopsis* under different stress conditions, *pGmbZIP60::GUS* transgenic *Arabidopsis* and WT were cultured on the same condition. Both groups were grown separately on 1/2 MS medium supplemented with 150 mM NaCl, 250 mM mannitol, 0.5 μM ABA, 150 μM MeJA, and 350 μM ETH. After 7 days, the seedlings were incubated in β-glucuronidase (GUS) staining solution at 37 °C overnight. Following this incubation, we used 75% ethanol to decolorize and observe the samples under a Leica M205 FA microscope.

We soaked approximately 100 rice seeds in a GA3 solution for 24 h for germination assays in rice. Seeds showing similar growth status were then selected and treated with aqueous solutions containing varying concentrations of NaCl (100 mM and 150 mM) and mannitol (250 mM and 350 mM) for 4 days. The bud lengths were measured using the Image J software, version 1.53. We cultivated seedlings in an 800 × Yoshida rice nutrient salt solution from Coolaber (Peking, China) for 15 days in the plant growth assays. After this period, seedlings showing similar growth status were transferred to a complete rice nutrient supplemented with 150 mM NaCl and 350 mM mannitol for 15 days. We used a Nikon camera to take photographs and Image J software to measure leaf and root lengths.

For the soybean growth assays, we cultivated approximately 36 soybean seeds for 15 days. We then selected seedlings showing similar growth status and treated them with aqueous solutions containing NaCl (150 mM) and mannitol (350 mM) for 15 days.

### 4.5. Quantitative Real-Time PCR

RNA was extracted from plant leaves for both the control and treatment groups. Reverse transcription was conducted using the All-in-one Super Mix for qPCR (TransGen, Beijing, China) to prepare templates for qPCR. The qPCR reaction system, with a total volume of 20 μL, comprised 10 μL of 2x TransStar Top Green qPCR SuperMix (TransGen, Beijing, China), 1 μL of cDNA, and 0.4 μL of gene-specific primers. The amplification conditions were as follows: 95 °C for 30 s, followed by 40 cycles of 94 °C for 5 s and 60 °C for 15 s. The actin gene served as the reference gene for qPCR analysis (Appendix A). Each reaction was performed in triplicate, along with three biological replicates. The results were quantitatively analyzed using the 2^–ΔΔCt^ method. In this study, a minimum of three biological replicates were used. Gene expression data were analyzed using one-way ANOVA in GraphPad Prism 9.0, with *p* < 0.05 considered statistically significant.

### 4.6. Chromatin Immunoprecipitation (ChIP) Analysis

Leaves from two-week-old transgenic soybean plants (*OE-2*) were cross-linked using a fixing solution [0.4 M sucrose, 10 mM Tris-HCl (pH = 8.0), 1% Triton X-100, 1 mM EDTA, 2% Formaldehyde, 0.5 mM β-Mercaptoethanol, 0.1 mM PMSF] [62]. After grinding the leaves in liquid nitrogen and cell lysate was added [45 mM HEPES (pH = 7.5), 135 mM NaCl, 0.9 mM EDTA, 1% Triton X-100, 10% glycerin, 0.1 mM PMSF, protease inhibitor cocktail (Roche)]. Chromosome fragmentation was carried out using 2 units of micrococcal nuclease (Sigma, St.Louis, MO, USA) in 1 mL of MNase digestion buffer [10 mM Tris-HCl (pH 8.0), 50 mM NaCl, 1 mM β-mercaptoethanol, 0.1% NP-40, 1 mM CaCl_2_, and protease inhibitor cocktail (Roche)]. The reaction was terminated with 5 mM EDTA. DNA fragments capable of binding to the GmbZIP60 were enriched using GFP antibodies (Abcam, Shanghai, China) and detected via qPCR. The details for the primer used are provided (Appendix A).

### 4.7. Detection of Photosynthetic Activity Fluorescence Parameters (FPAD)

A handheld photosynthetic activity fluorescence parameter detector (SPAD-502Plus, KONICA MINOLTA, Tokyo, Japan) measured different leaves on the same plant. The measured values represent the photosynthetic activity fluorescence parameters (FPAD) of a 2 mm × 3 mm area on the leaves of living plants in the morning, which can indicate the photosynthetic capacity of the plant’s leaves.

### 4.8. Detection of Leaf Water Content

Weigh a clean centrifuge tube. Place several fresh leaves into the clean centrifuge tube, weigh it again, and cover the tube with a sealing film with holes. Place the tube into the oven and dry it until a constant weight is achieved. Weigh it again. Calculate the *water* content:$$\frac{\left(Fresh\;weight-Dry\;weight\right)}{Fresh\;weight}\times 100\%$$

## 5. Conclusions

In this study, we identified a soybean bZIP gene, *GmbZIP60*. Our results showed that overexpression of *GmbZIP60* will increase resistance to tolerance of abiotic stresses by regulating phytohormone-responsive genes and abiotic stress-responsive genes. These findings enhance the understanding of the role of soybean *GmbZIP60* transcription factor in the complex abiotic stress molecular mechanisms, and served as a theoretical foundation for enhancing the stress tolerance of soybean.

## Figures and Tables

**Figure 1 ijms-26-03455-f001:**
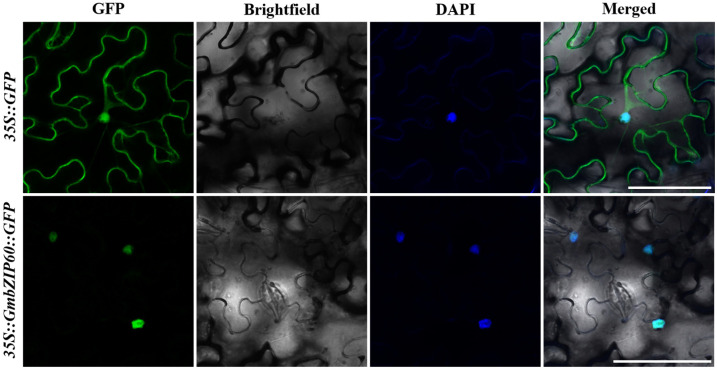
Subcellular localization of GmbZIP60 in tobacco. *35S::GFP* and *35S::GmbZIP60::GFP* were transiently expressed in the leave cells of *Nicotiana benthamiana* and observed using a laser scanning confocal microscope. Scale bar = 100 μm.

**Figure 2 ijms-26-03455-f002:**
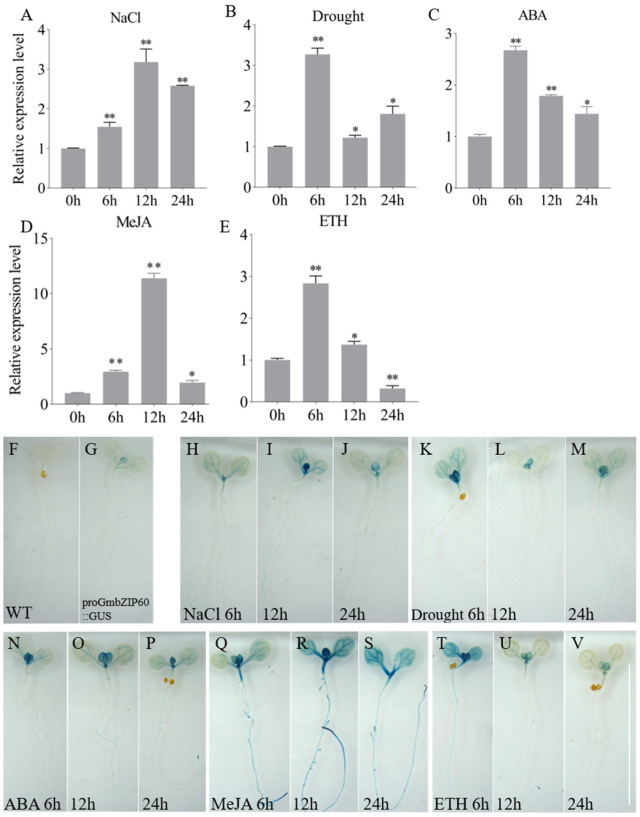
Expression patterns and GUS histochemical staining of *GmbZIP60* under abiotic stresses. The two-week-old soybean seedings were treated with NaCl (**A**), drought (**B**), ABA (**C**), MeJA (**D**), and ETH (**E**). GUS staining results are shown for untreated seedlings (**F**,**G**), and seedlings treated withNaCl (**H**–**J**), drought (**K**–**M**), ABA (**N**–**P**), MeJA (**Q**–**S**), and ETH (**T**–**V**). Scale bar = 1 cm. Errors bars represent ± SD of three biological replicates. Asterisks indicate significant differences for the indicated comparisons based on a Student’s *t-*test (** *p* < 0.01; 0.01 < * *p* < 0.05).

**Figure 3 ijms-26-03455-f003:**
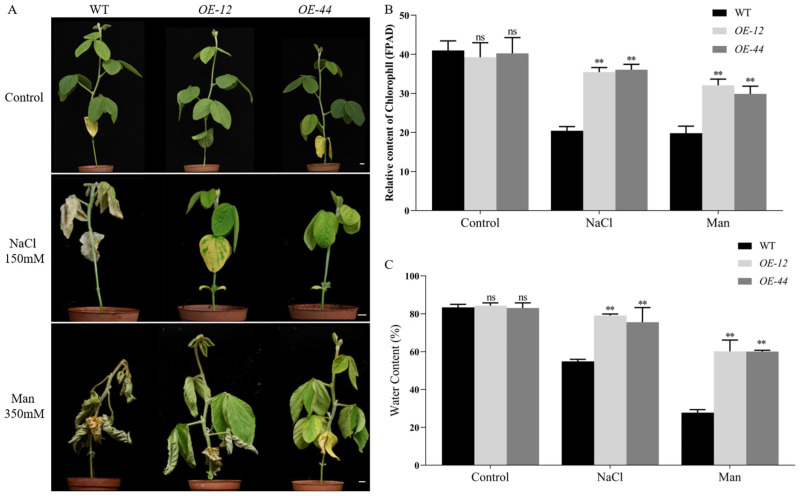
Phenotypic analysis of *OE-GmbZIP60* transgenic soybean plants in response to under salt and drought treatment stress. Two-week-old seedlings were grown under control conditions or supplemented with 150 mM NaCl and 350 mM mannitol for 20 days. *OE-GmbZIP60* transgenic soybean plants under salt and drought stress were analyzed for phenotypic (**A**), photosynthetic activity fluorescence parameters (**B**), and water content (**C**), Scale bar, 1 cm. Errors bars represent ± SD of three biological replicates. Asterisks indicate significant differences for the indicated comparisons based on a Student’s *t-*test (** *p* < 0.01).

**Figure 4 ijms-26-03455-f004:**
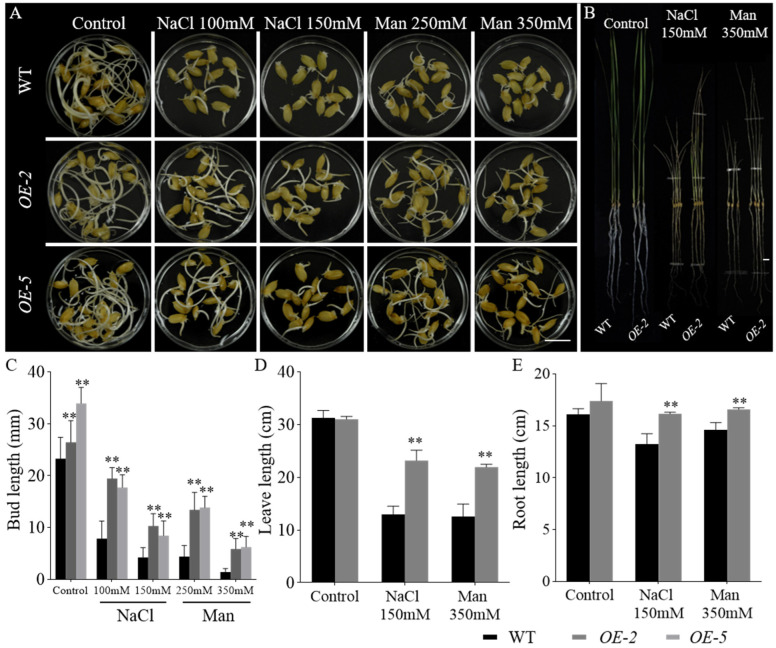
Phenotypic analysis of *OE-GmbZIP60* transgenic rice plants under salt and drought stress. (**A**) Seeds were germinated under control conditions or subjected to 100 mM and 150 mM NaCl and 250 mM and 350 mM mannitol treatments for 4 days. Scale bar, 1 cm. (**B**) Plants were germinated under control or exposed to 150 mM NaCl and 350 mM mannitol for 10 days. Scale bar, 1 cm. (**C**) Measurement of the seedling bud length. (**D**) Measurement of the plant leaf length. (**E**) Measurement of the plant root length. Errors bars represent ± SD of three biological replicates. Asterisks denote significant differences between the indicated comparisons based on a Student’s *t-*test (** *p* < 0.01).

**Figure 5 ijms-26-03455-f005:**
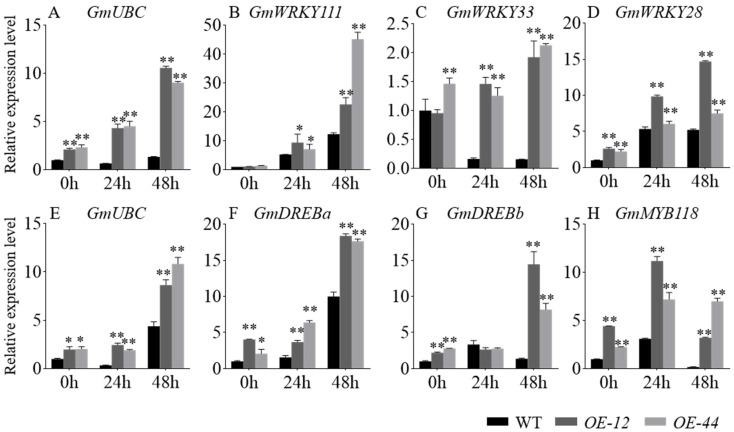
Expression of abiotic stress-related genes in the WT and *OE-GmbZIP60* transgenic soybean plants in response to salt (**A**–**D**) and drought (**E**–**H**) stress. Statistical results are provided in Appendix A. Errors bars indicate ± SD of three biological replicates. Asterisks indicate significant differences for the indicated comparisons based on a Student’s *t-*test (** *p* < 0.01; 0.01 < * *p* < 0.05).

**Figure 6 ijms-26-03455-f006:**
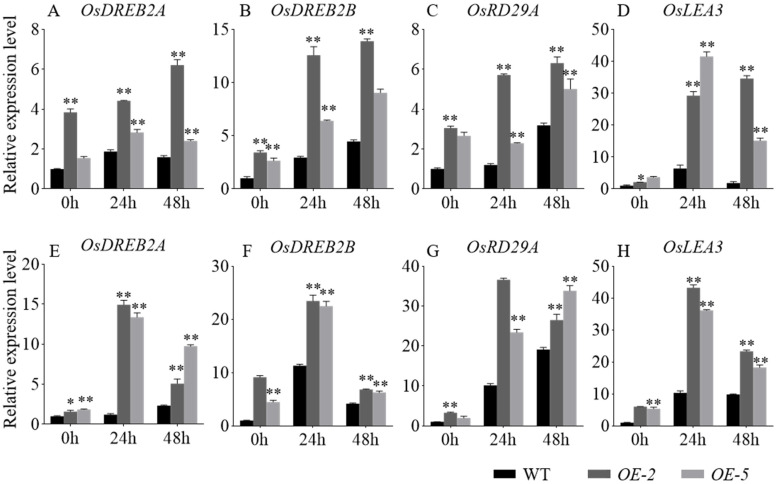
Expression of abiotic stress-related genes in the WT and *OE-GmbZIP60* transgenic rice plants in response to salt (**A**–**D**) and drought (**E**–**H**) stress. Statistical results are provided in Appendix A. Errors bars indicate ± SD of three biological replicates. Asterisks indicate significant differences for the indicated comparisons based on a Student’s *t-*test (** *p* < 0.01; 0.01 < * *p* < 0.05).

**Figure 7 ijms-26-03455-f007:**
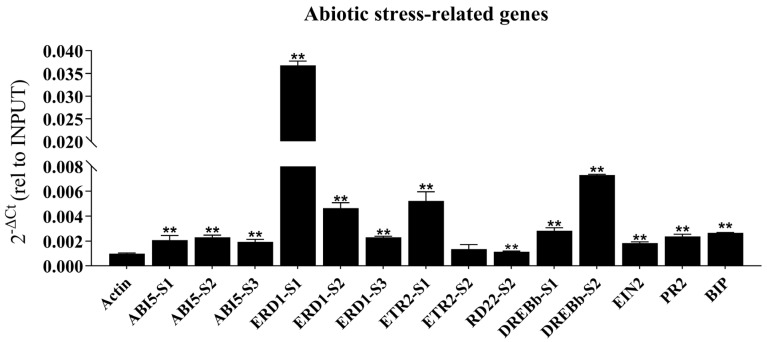
Chromatin immunoprecipitation (ChIP) analysis of *OE-GmbZIP60* transgenic soybean plants. ChIP-qPCR analysis showing GmbZIP60 binding to the promoters of abiotic stress-related genes in *OE-GmbZIP60* transgenic soybean plants, using a GFP antibody. Errors bars indicate ± SD of three biological replicates (S represents the existence of different binding sites). Asterisks indicate significant differences for the indicated comparisons based on a Student’s *t-*test (** *p* < 0.01).

**Figure 8 ijms-26-03455-f008:**
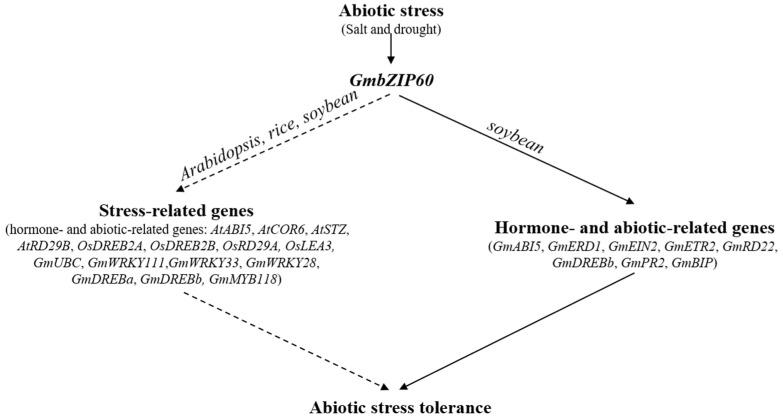
A schematic model of *GmbZIP60* mediated abiotic stress tolerance in transgenic *Arabidopsis*, rice, and soybean. *GmbZIP60* positively modulates the abiotic stress tolerance: *GmbZIP60* positively regulates the expression of hormone and abiotic-related genes. The dashed lines indicate indirect regulation, and solid lines indicate direct regulation. The arrows indicate induction or positive modulation. This figure was adapted from [41].

## Data Availability

All data analyzed during this study are included in this article and Appendix A.

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
