# Peer review of "Overexpression of the Transcription Factor GmbZIP60 Increases Salt and Drought Tolerance in Soybean (Glycine max)"

_ijms, 2025, doi:10.3390/ijms26073455_

Round 1
Reviewer 1 Report
Comments and Suggestions for Authors
This article investigates the role of the transcription factor GmbZIP60 in enhancing salt and drought tolerance in soybean (Glycine max) and rice (Oryza sativa). The study employs various techniques, including gene expression analysis, transgenic plant phenotype observations, and chromatin immunoprecipitation qPCR (ChIP-qPCR), to demonstrate the potential of GmbZIP60 in these crops. The experimental design is sound, and the data support the research hypothesis well. However, there are several areas that could be further improved to enhance the academic quality and reproducibility of the study.
1.The study focuses on the GmbZIP60 gene in soybean (Glycine max), although overexpression experiments were also conducted in rice (Oryza sativa) to validate the function. Since the primary focus of the research is on soybean, I suggest revising the title to better reflect this focus. The current title may give the impression that both soybean and rice are treated equally, while rice is primarily used for validation purposes.
- Introduction. Can the authors provide a more detailed comparison of GmbZIP60 to other bZIP transcription factors that also respond to abiotic stress in plants?
- Provide more insight into the rationale behind the choice of plant hormones (ABA, ETH, MeJA) used in the study? Specifically, are these treatments considered part of the biotic or abiotic stress response pathways? How do these hormone treatments relate to previous studies in the literature, particularly in terms of their known roles in plant stress responses?
- Are there physiological data (e.g., leaf water content, photosynthesis rate) that correlate with the observed molecular results in transgenic plants?
- Are there any significant differences in growth or yield between the transgenic and wild-type plants under non-stress conditions?
- the authors speculate on how GmbZIP60 interacts with other transcription factors and regulatory networks under abiotic stress conditions?
- Figure 1, DAPI Exposure: The DAPI exposure in Figure 1 seems too strong, affecting the overall quality of the image. I suggest adjusting the exposure settings to ensure clearer and more balanced staining of the cell nuclei without overexposing the image.8.
8.Figure 2, In Figure 2A, the time point "24h" is missing the unit "h" (hours). Please add the time unit for clarity. Additionally, I recommend providing a higher resolution version of this image to ensure that the details are more clearly visible.
9.Figure 7, Font Resolution: The font resolution in Figure 7 appears to be insufficient, making it difficult to read. Please consider increasing the resolution of the font or using a clearer, higher-quality font for better readability.
- There are several grammatical issues throughout the manuscript. For example, on lines 403-405 in the conclusion section, the authors state: "These finding enhance the understanding of the role of soybean GmbZIP60 transcription factor in the complex abiotic stress molecular mechanisms." However, the word "finding" should be in its plural form.
Comments on the Quality of English Language
The English could be improved to more clearly express the research.
Author Response
Q1: The study focuses on the GmbZIP60 gene in soybean (Glycine max), although overexpression experiments were also conducted in rice (Oryza sativa) to validate the function. Since the primary focus of the research is on soybean, I suggest revising the title to better reflect this focus. The current title may give the impression that both soybean and rice are treated equally, while rice is primarily used for validation purposes.
Response 1: Thank you very much for your insightful comment. We have revised the title to “Overexpression of the Transcription Factor GmbZIP60 Increases Salt and Drought Tolerance in Soybean (Glycine max)”.
Q2: Introduction. Can the authors provide a more detailed comparison of GmbZIP60 to other bZIP transcription factors that also respond to abiotic stress in plants?
Response 2: Thank you very much for your valuable suggestion. We appreciate your interest in a more detailed comparison of GmbZIP60 with other bZIP transcription factors involved in abiotic stress responses. In response to your comment, we have revised the Introduction section to provide a more comprehensive comparison. The added text can be found on page 3, lines 72-98.
Q3: Provide more insight into the rationale behind the choice of plant hormones (ABA, ETH, MeJA) used in the study. Specifically, are these treatments considered part of the biotic or abiotic stress response pathways? How do these hormone treatments relate to previous studies in the literature, particularly in terms of their known roles in plant stress responses?
Response 3: We have incorporated additional information to provide a more comprehensive explanation that can be found on page 7, lines 225-234. We hope this addition clarifies the rationale behind our choice of these hormones and how they relate to both biotic and abiotic stress response pathways.
Q4: Are there physiological data (e.g., leaf water content, photosynthesis rate) that correlate with the observed molecular results in transgenic plants?
Response 4: Thank you for your insightful question regarding the correlation between molecular results and physiological data in transgenic plants. We have now included additional experimental results to address this point. Specifically, we have added data on leaf water content and photosynthetic activity fluorescence parameters, which provide physiological evidence supporting the observed molecular changes in the transgenic plants. The detailed results can be found on page 6, lines 184-193, and in Figure 6 on page 3.
Q5: Are there any significant differences in growth or yield between the transgenic and wild-type plants under non-stress conditions?
Response 5: Thank you for your question regarding the growth and yield differences between transgenic and wild-type plants under non-stress conditions. We have added the following information to address this point,and his content can be found on page 6, lines 178-180.
Q6: the authors speculate on how GmbZIP60 interacts with other transcription factors and regulatory networks under abiotic stress conditions?
Response 6: Thank you for your insightful question regarding the potential interactions of GmbZIP60 with other transcription factors and regulatory networks under abiotic stress conditions. We have expanded our discussion to provide more context on this topic. Specifically, we have added the following text that can be found on page 13, lines 376-383.
Q7: Figure 1, DAPI Exposure: The DAPI exposure in Figure 1 seems too strong, affecting the overall quality of the image. I suggest adjusting the exposure settings to ensure clearer and more balanced staining of the cell nuclei without overexposing the image.8.
Response 7: Thank you for your comment regarding the DAPI exposure in Figure 1. We have carefully adjusted the exposure settings to achieve clearer and more balanced staining of the cell nuclei, ensuring that the image quality is improved without overexposure. The revised figure can be found on page 4, Figure 1.
Q8: Figure 2, In Figure 2A, the time point "24h" is missing the unit "h" (hours). Please add the time unit for clarity. Additionally, I recommend providing a higher resolution version of this image to ensure that the details are more clearly visible.
Response 8: Thank you for your careful review and suggestions regarding Figure 2. We have made the following revisions to address your concerns: The missing unit "h" (hours) has been added to the "24h" time point in Figure 2A to ensure clarity. Image Resolution: We have also provided a higher-resolution version of Figure 2A to ensure that the details are more clearly visible. The revised figure can be found on page 6, Figure 2.
Q9: Figure 7, Font Resolution: The font resolution in Figure 7 appears to be insufficient, making it difficult to read. Please consider increasing the resolution of the font or using a clearer, higher-quality font for better readability.
Response 9: Thank you for your feedback regarding the font resolution in Figure 7. We have addressed this issue by increasing the font resolution and ensuring that the text is clearer and more legible. The revised figure now uses a higher-quality font to enhance readability. The updated Figure 7 can be found on page 12.
Q10: There are several grammatical issues throughout the manuscript. For example, on lines 403-405 in the conclusion section, the authors state: "These finding enhance the understanding of the role of soybean GmbZIP60 transcription factor in the complex abiotic stress molecular mechanisms." However, the word "finding" should be in its plural form.
Response 10: Thank you for pointing out the grammatical issues in our manuscript. We apologize for any confusion caused by these errors. We have carefully reviewed the text and made the necessary corrections to ensure grammatical accuracy. Specifically, we have corrected the sentence in the conclusion section on page 17, lines 541-544.
Reviewer 2 Report
Comments and Suggestions for Authors
In this manuscript, the authors cloned one of the bZIP proteins in soybean, GmbZIP60. The authors first verified the nucleus expression of GmbZIP60. Then the authors characterized the expression patterns of GmbZIP60 in Arabidopsis under salt or drought stress and various hormone treatment. The authors also generated GmbZIP60 overexpression strains of soybean and rice and found that these strains are more resistant to salt and drought stress and also more sensitive to hormone treatment. The authors also used qRT-PCR to verified that the expression level of several genes known to be involved in abiotic stress response it higher than that of WT under salt and drought stress. The authors also used CHIP-qPCR to identify several targets proteins of GmbZIP60 in soybean.
Overall the experimental design is solid. The presentation of data is mostly clear and the data supports the authors’ conclusions.
The reviewer points out a few points that need correction or further clarification.
- Line 108, “increased gradually” does not agree with the data. The data showed gradual decrease.
- In Figure S2 and text, one of the overexpression strains of both Arabidopsis and soybean are both named OE-44 and it may cause confusion despite the authors clarification in the figure legend. The reviewer suggests that the authors change the name of at least one of the strains.
- In Figure 3, the authors showed that the OE strains are more resistant to salt stress that WT and showed representative pictures of the plant. Although in the picture, the WT is much more wilted than the OE strains, it would be better to have some kind of quantification and statistcs to summarize the behavior of different repeats of the OE strain and the WT. One more question about this figure is that the OE-44 strain seems to be more resistant that the OE-12 strain judged by less wilting and this is counterintuitive to the data in figure S2 which shows that the expression level of bZIP60 is about 2 times higher in OE-12 than in OE-44. One possibility is that there is some dose-related relationship between bZIP60 expression and salt resistance which the authors have yet characterized. Alternatively, this could be because that the authors only showed the picture of one representative plant. This would further demonstrate the need of better quantification of the wilting in this figure.
- In Figure 7, some of the genes are labeled as “ERD1-1, -2, -3”. However, it is not clear from the manuscript or figure legends what these stand for. Are these different repeats? If so, why some of the other genes do not have the repeat? And why ERD1-1 is way higher than ERD1-2 or ERD1-3?
- In the introduction, the authors provided much information on bZIP proteins in general. However, the authors did not provide much information specifically on GmbZIP60 and why among many bZIP proteins GmbZIP60 is special and becomes the protein of interest. Providing more information on this topic could strengthen the rationale of this study and the significance and novelty of the manuscript.
- In the author affiliation part 2, “Guangi” should be “Guangxi”.
Comments on the Quality of English Language
There are minor grammatical errors
Author Response
Q1. Line 108, “increased gradually” does not agree with the data. The data showed gradual decrease.
Response 1: Thank you for pointing out the inconsistency on line 108. We apologize for the oversight and appreciate your attention to detail. We have revised the sentence to accurately reflect the data. This revision can be found on page 4, line 146-153.
Q2. In Figure S2 and text, one of the overexpression strains of both Arabidopsis and soybean are both named OE-44 and it may cause confusion despite the authors clarification in the figure legend. The reviewer suggests that the authors change the name of at least one of the strains.
Response 2: Thank you for raising this critical point regarding naming the overexpression strains in Figure S2 and the text. We acknowledge that using the same strain name (OE-44) for both Arabidopsis and soybean could potentially confuse, despite the clarification provided in the figure legend. To address this issue, we have renamed the Arabidopsis overexpression strain from OE-44 to OE-18 in both the main text and Figure S2. The updated information can be found in the relevant sections of the manuscript and in Figure S2.
Q3. In Figure 3, the authors showed that the OE strains are more resistant to salt stress that WT and showed representative pictures of the plant. Although in the picture, the WT is much more wilted than the OE strains, it would be better to have some kind of quantification and statistics to summarize the behavior of different repeats of the OE strain and the WT. One more question about this figure is that the OE-44 strain seems to be more resistant that the OE-12 strain judged by less wilting and this is counterintuitive to the data in figure S2 which shows that the expression level of bZIP60 is about 2 times higher in OE-12 than in OE-44. One possibility is that there is some dose-related relationship between bZIP60 expression and salt resistance which the authors have yet characterized. Alternatively, this could be because that the authors only showed the picture of one representative plant. This would further demonstrate the need of better quantification of the wilting in this figure.
Response 3: Thank you for your detailed and constructive feedback regarding Figure 3. We appreciate your suggestions for improving the quantification and statistical analysis of our results. In response to your comments, we have supplemented the manuscript with additional data to better quantify the differences between the OE strains and the wild-type (WT) plants under salt stress conditions. Specifically, we have included measurements of relative chlorophyll and water content, providing a more comprehensive assessment of plant health and stress tolerance. These new data help to quantify the observed differences in wilting and overall plant condition. The additional data can be found on page 7, lines 225-234, and in Figure 3 on page 8. We observed gene expression variations among overexpression lines under non-stress conditions. Under drought and salt stress, different plants showed varying stress responses, suggesting differences in GmbZIP60 expression between OE-12 and OE-44. Although qRT-PCR showed higher GmbZIP60 expression in OE-12, protein levels may have reached a threshold, making OE-12 and OE-44 comparable. We have added an explanation in the discussion, which can be found on page 12, line 349-357.
Q4. In Figure 7, some of the genes are labeled as “ERD1-1, -2, -3”. However, it is not clear from the manuscript or figure legends what these stand for. Are these different repeats? If so, why some of the other genes do not have the repeat? And why ERD1-1 is way higher than ERD1-2 or ERD1-3?
Response 4: Thank you for your insightful questions regarding the labeling of genes in Figure 7. We apologize for any confusion this may have caused. We have clarified this in both the main text and the figure legend to ensure that readers understand the significance of these multiple binding sites. The revised information can be found on page 12, lines 324-325, and in the legend of Figure 7 on page 12.
Q5. In the introduction, the authors provided much information on bZIP proteins in general. However, the authors did not provide much information specifically on GmbZIP60 and why among many bZIP proteins GmbZIP60 is special and becomes the protein of interest. Providing more information on this topic could strengthen the rationale of this study and the significance and novelty of the manuscript.
Response 5: Thank you for your insightful comments regarding the introduction section of our manuscript. We appreciate your suggestion to provide more specific information on GmbZIP60 and its unique significance among the bZIP protein family. In response to your feedback, we have expanded the introduction to include additional details that highlight why GmbZIP60 was specifically chosen as the protein of interest in this study. We have added a section that explains the unique characteristics and functions of GmbZIP60, distinguishing it from other members of the bZIP family. This additional information underscores the rationale and novelty of our research. The revised introduction can be found on page 3, lines 94-98.
Q6. In the author affiliation part 2, “Guangi” should be “Guangxi”.
Response 6: Thank you for pointing out the typographical error in the author affiliation section. We apologize for the oversight and appreciate your attention to detail. We have corrected the mistake, and "Guangi" has been changed to "Guangxi" in the affiliation details.